# Circulating 20S proteasome for assessing protein energy wasting syndrome in hemodialysis patients

Julien Aniort[1,2☯]*, Marine Freist[1,3☯], Aurélien Piraud[1], Carole Philipponnet[1], Mohamed Hadj Abdelkader[1], Cyril Garrouste[1], Elodie Gentes[4], Bruno Pereira[5], Anne-Elisabeth Heng[1,2]

**1** Nephrology, Dialysis and Transplantation Department, Gabriel Montpied University Hospital, Clermont-Ferrand, France, **2** INRA, UMR 1019, Human Nutrition Unit (UNH), St Genès Champanelle, France, **3** Nephrology and Dialysis Department, Emile Roux Hospital, Le Puy en Velay, France, **4** Clinical Nutrition Department, Gabriel Montpied University Hospital, Clermont-Ferrand, France, **5** University Hospital of Clermont-Ferrand, Biostatistics unit (DRCI), Clermont-Ferrand, France

☯ These authors contributed equally to this work.
* janiort@chu-clermontferrand.fr

**Data Availability Statement:** All relevant data are within the paper and its Supporting Information files.

## Abstract

Protein energy wasting (PEW) including muscle atrophy is a common complication in chronic hemodialysis patients. The ubiquitin proteasome system (UPS) is the main proteolytic system causing muscle atrophy in chronic kidney disease and proteasome 20S is the catalytic component of the UPS. Circulating proteasome 20S (c20S proteasome) is present in the blood and its level is related to disease severity and prognosis in several disorders. We hypothesized that c20S proteasome could be related with muscle mass, other PEW criteria and their evolution in hemodialysis patients. Stable hemodialysis patients treated at our center for more than 3 months were followed over 2 years. C20S proteasome assay was performed at baseline. Biological and clinical data were collected, muscle mass was assessed by multi-frequency bio-impedancemetry, and nutritional scores were calculated at baseline, 1 year and 2 years. Hospitalizations and mortality data were collected over the 2 years. Forty-nine patients were included. At baseline, the c20S proteasome level was 0.40 [0.26–0.55] μg/ml. Low muscle mass as defined by a lean tissue index (LTI) < 10th in accordance with the International Society of Renal Nutrition and Metabolism guidelines was observed in 36% and PEW in 62%. Increased c20S proteasome levels were related with LTI at baseline (R = 0.43, p = 0.004) and with its 2 year-variation (R = -0.56, p = 0.003). Two-year survival rate was not different between higher and lower c20S proteasome values (78.9 vs 78.4%, p = 0.98 log-rank test). C20S proteasome is not a good marker for assessing nutritional status in hemodialysis patients and predicting patient outcomes.

## Introduction

The protein energy wasting (PEW) syndrome, including muscle atrophy, is frequently encountered in hemodialysis patients and contributes to the high mortality rate observed in this

**Funding:** The authors received no specific funding for this work.

**Competing interests:** The authors have declared that no competing interests exist.

population [1]. It can be caused by a decrease in energy and protein intakes, sedentariness, hormonal disorders, metabolic acidosis and inflammation. The decrease in muscle mass is due to an imbalance in protein synthesis and proteolysis in favor of the latter [2]. The ubiquitin-proteasome proteolytic system (UPS) is crucial for the atrophying process, notably in chronic kidney disease (CKD) [3, 4]. The first step in myofibrillar protein degradation is thought to be activation of caspase 3 [5]. The UPS then targets the proteins to be degraded, (such as myosin [6] and actin [7], by linking covalently a ubiquitin (Ub) chain. Ub chain formation is catalyzed by an enzymatic cascade (E1, E2, E3), with E3s that recognize the substrates. The 26S proteasome consists of a core particle with proteolytic activity (proteasome 20S) and two regulator particles (proteasome 19S) at its extremities [8]. The 26S proteasome recognizes and degrades polyubiquitinated proteins. In animal models of CKD-induced muscle atrophy, studies of muscle biopsies have shown an increase in the 14kDa actin level, a "footprint" of caspase 3 activity [5], increased expression of E3 ligases such as the muscle- specific E3 ligases MuRF1 and MAFbx [9, 10], and proteasome subunits[11]. In humans, activation of the UPS is evidenced by an increased expression of several E3 ligases in the muscle of hemodialysis patients [4]. However, it is not possible to perform a muscle biopsy in all patients to detect the activation of UPS. This is why the identification of biomarkers associated with the onset of muscle wasting and that are easily measurable in the blood would be more useful. 20S proteasome measured in plasma by ELISA seems a good candidate for this assessment. Increases in circulating 20S proteasome (c20S proteasome) have been widely reported in hematologic malignancies, solid cancers, burn injury, sepsis and autoimmune diseases. c20S proteasome levels are associated with poor prognosis and increased mortality in this setting [12]. To date only one study, by *Fukasawa et al.*, has looked at the c20S proteasome as a marker of muscle atrophy. The authors found a weak but significant correlation in hemodialysis patients between c20S proteasome, abdominal muscle area measured by computed tomography and creatinine production [13]. Thus, we hypothesized that an increase in the c20S proteasome level could predict the evolution of PEW parameters and patient outcomes. The aim of this study was to determine whether a c20S proteasome assay is predictive of muscle mass loss and PEW in hemodialysis patients.

## Material and method

### Study design and subjects

We conducted a prospective cohort study at the dialysis center of the university hospital of Clermont-Ferrand (France). Patients over 18 years old and treated by hemodialysis or hemodiafiltration for more than 3 months were selected. Exclusion criteria were lack of consent, acute infection or inflammatory disease during the 3 months before inclusion, serum C reactive protein level (CRP) > 30 mg/L, active malignancy disease, hypercortisolism, uncontrolled dysthyroidism and neuromuscular pathology. Patients were followed from March 2016 to March 2018 or death or until transplantation, transfer to another center or the end of follow-up (march 20). The study protocol has been reviewed and approved by the local research ethic committee, (CPP Sud-Est VI, no. 2016-AU1246) and the French national agency for the safety of medicines and health products (IDRCB: 2016-A00204-47). All patients provided written informed consent before participation.

### Biochemical and clinical measurements

Demographic and clinical data (gender, age, cause of end stage renal diseases, dialysis vintage, technique and comorbidities) were obtained at baseline from medical records. The date and cause of death, number of hospitalizations, dates and causes of admission were collected over 2

years. Blood samples were drawn at the start and the end of the middle week dialysis sessions at baseline, and after 1 and 2 years of follow-up. Urea and creatinine serum levels before and after dialysis sessions, albumin, prealbumin, bicarbonate, hemoglobin, CRP, parathyroid hormone (PTH), vitamin D, iron, ferritin and transferin serum levels were measured by standard laboratory techniques with an auto-analyzer.

### Anthropometric measurements

Anthropometric measurements (body weight, height, mid-arm circumference, triceps skinfold thickness) were performed after a mid-week dialysis session at inclusion and at 1 year and 2 years. Body mass index (BMI) was calculated with the ratio body weight /size$^2$ (kg/m$^2$) and mid-arm muscle circumference (MAMC) with the following formula: mid-arm circumference– 3.142 x triceps skinfold thickness (cm).

### Bioimpedancemetry

Body composition [adipose tissue mass (ATM), lean tissue mass (LTM), fat tissue Index (FTI), lean tissue index (LTI) and overhydration (OW)] was evaluated at inclusion, 1 year and 2 years by multifrequence bioimpedancemetry using a Body Composition Monitor (BCM ® Fresenius Medical Care, Bad Homburg, Germany). Measurements were performed after 10 minutes of rest in the lying position, before the start of a mid-week dialysis session.

### Proteasome measurement

C20S proteasome was measured by enzyme-linked immunosorbent assay (ELISA) using the proteasome ELISA kit (Enzo Life Sciences, Farmingdale, New York, United-States) at inclusion and at the end of follow-up. As previously reported [13], 5 ml of blood were drawn before a mid-week dialysis session in a dry tube with separating gel and then centrifuged. Aliquots were then prepared and stored at -20˚C. The ELISA plate is coated with the capture antibody. The detection antibody is then added, followed by the secondary antibody conjugated with horseradish peroxidase. Peroxydase reacts with TMB (3, 3', 5, 5'-tetramethylbenzidine) substrate to produce a blue solution. The color intensity at 450 nm is directly proportional to proteasome concentration.

### Formula and calculated scores

Kt/V$_{sp}$ was calculated with the formula of *Daugirdas et al* to evaluate dialysis dose: Kt/V$_{sp}$ = ln (R—0.008 × t) + (4–3.5 × R) × 0.55 UF/W, where R is the ratio of postdialysis to predialysis urea level, UF the ultrafiltration volume and W the post-dialysis body weight [14]. Normalized protein catabolic rate (nPCR) was calculated according to the model of *Depner et al*: nPCR (g/kg/day) = (Urea × 2.801) / (25.8 + 1.15 Kt/V$_{sp}$ + 56.4/ Kt/V$_{sp}$) + 0.168. Urea is serum urea (mmol/L) before dialysis [15]. Creatinine index was calculated according to *Canaud et al* with the following formula: creatinine index (mg/kg/day): 16.21 + 1.12 × [1 (male); 0 (female)]-0.06 × age—0.08 × Kt/V$_{sp}$ + 0.009 × creatinine, age in years, creatinine in μmol/L [16]. The Prognostic Inflammatory and Nutritional Index (PINI) was calculated by the following formula: PINI = (alpha1-Acid Glycoprotein (α-AG) = orosomucoid in mg/L × CRP mg/L) / (albumin in g/L × prealbumin in mg/L)[17]. Protein energy wasting syndrome (PEW) was defined according to guidelines of the International Society of Renal Nutrition and Metabolism (ISRNM) [18]. However, energy and protein intakes are not routinely collected in our center and nPCR was used to assess protein intake. A simplified definition of PEW was used [19] based on fewer than three of the following items: albumin > 35g/dl, BMI > 23kg/m$^2$,

serum creatinine/body surface area (BSA) > 380 µmol/l/m$^2$ (with BSA estimated using Boyd formula) and nPCR > 0.8g/kg/day.

## Statistical analysis

All statistical analyses were performed with Stata software (version 13, Statacorp, College Station, US). The tests were two-sided with a type-I error set at 5%. Categorical data were presented as numbers and percentages and continuous data as mean ± standard-deviation (SD) or median (interquartile range) according to the statistical distribution. The assumption of normality was assessed by the Shapiro-Wilk test. Univariate analyses were then carried out to compare variables between independent groups using Student t-test or Mann-Whitney test when the hypotheses of t-test were not met for continuous parameters and Chi2 test or, if applicable, Fisher's exact test for categorical variables. The study of relations between continuous variables was analyzed estimating correlation coefficient, Pearson or Spearman according to the statistical distribution and applying a Sidak's type I error correction to take into account multiple comparisons. Survival was estimated by the Kaplan-Meier method. To identify prognostic predictors, comparisons were performed with the log-rank test in univariate analyses and Cox proportional-hazards regression. The proportional-hazard hypothesis was tested by Schoenfeld's test and plotting residuals.

## Results

### Baseline characteristics

At the start of the study, 96 patients were being treated in our center, of whom 49 were enrolled and followed. A flow chart is provided in Fig 1.

Baseline characteristic of the patients are shown in Table 1. The cause of ESRD was chronic glomerulonephritis (35%), vascular nephropathy (18%), polycystic kidney disease (10%), diabetic nephropathy (4%) and other/unknown (35%). Four (8%) patients were being treated with perdialytic parenteral nutrition and 21 (45%) with oral nutritional supplements. Three (6%) patients had Kt/V$_{sp}$ < 1.2, 6 (12%) patients had a predialysis bicarbonate blood level below 22 mmol/L, 9 (18%) a PTH level above nine times the upper limit of normal value, and 18 (37%) had a 25OH Vitamin D3 < 30 µg/L.

Values of nutritional markers at inclusion are shown in Table 2. BMI < 23 was observed in 13 (27%) patients. Patients had a serum albumin < 35 g/L or a serum prealbumin < 0.30 g/L in 41% and 47% of cases, respectively. Low muscle mass as defined by a LTI < 10th was observed in 16 (33%) patients. PINI > 1 was recorded in 15 (31%) patients, and PEW in 24 (49%) and 30 (61%) patients according to ISRNM or simplified definition, respectively.

### c20S proteasome and correlation with nutritional markers at baseline and during follow-up

At inclusion, c20S proteasome level were 0.40[0.26–0.55] µg/ml. Pearson correlation coefficients between c20S proteasome levels and different nutritional parameters are resumed in Table 3. Increased c20S proteasome level were positively related with LTI at baseline and negatively with 2 years LTI variation (Fig 2). C20S proteasome was not different in patients with PEW at baseline according ISRNM (0.42 ± 0.21 Vs 0.45 ± 0.19, p = 0.73) or according the simplified definition (0.39 ±0.19 Vs 0.51 ±0.22, p = 0.21) in comparison of patients without PEW.

### c20S proteasome levels and patients outcomes

During the follow-up period 11 patients died (22.9%), 3 patients died during the first year and 8 during the second year of follow-up. The causes of death were cardiac disease (n = 2),

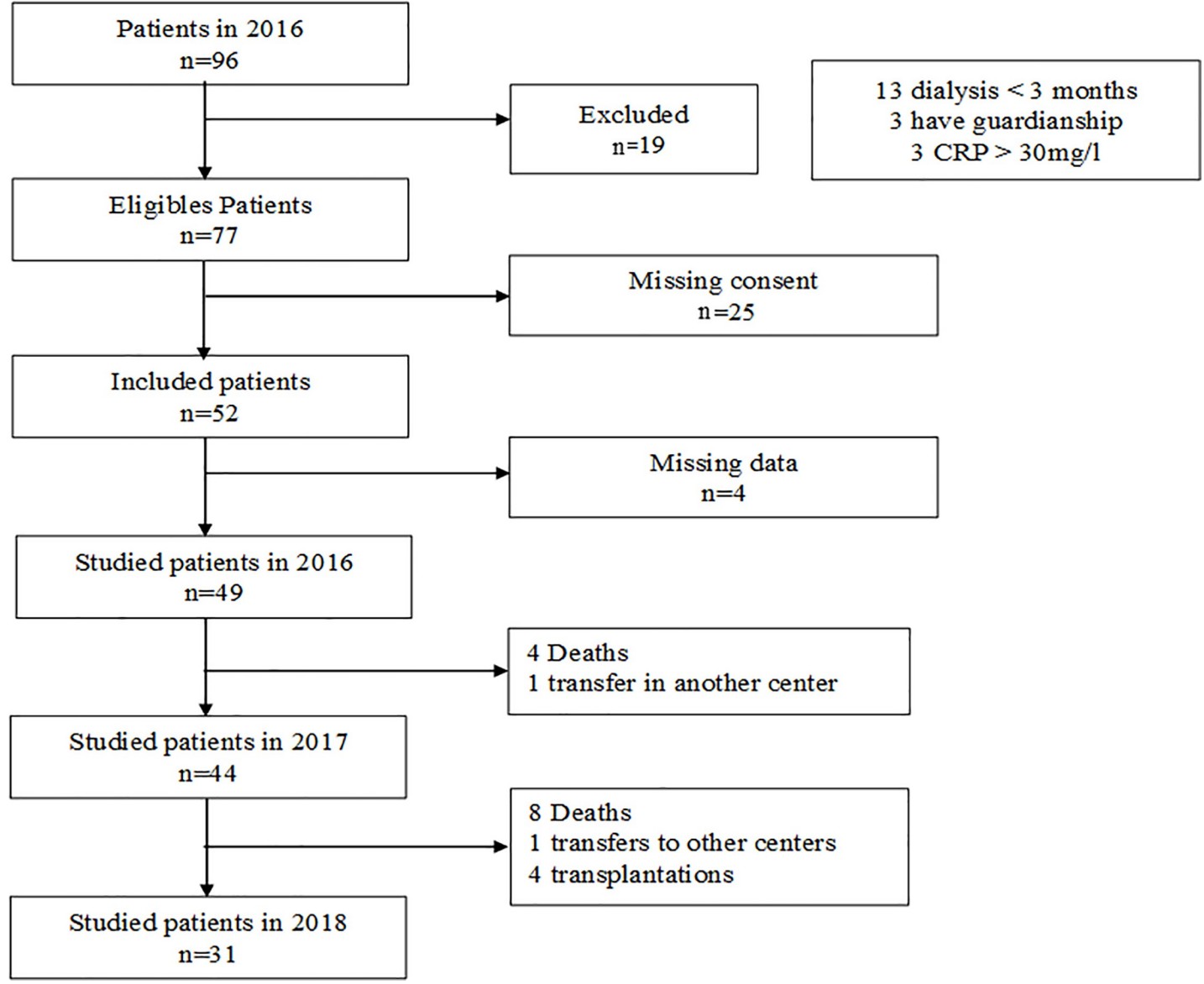

**Fig 1. Flow chart.** CRP, C-reactive protein.

cerebral stroke (n = 1), neoplasia (n = 1), pulmonary embolism (n = 1), calciphylaxis (n = 1), hemorrhage (n = 1), hemodialysis withdrawal (n = 2) and unknown (n = 1). C20S proteasome level (HR = 1.124, 95%CI [0.05–24.8], p = 0.94) and a proteasome level above the median value (HR = 0.75, 95%CI [0.21–2.59], p = 0.63) did not predict patient survival at two years. Among nutritional markers, only a prealbumin level < 30 g/L was associated with decreased patient survival at two years (HR = 5.1, 95%CI[1.10–24.1], p = 0.03) (Fig 3). During the follow-up, 28 patients were hospitalized. Admission rates did not differ between patients having above or under the median c20S proteasome value (p = 0.15).

## Discussion

This study assessed the value of the c20S proteasome in the assessment of nutritional status, prediction of its evolution and its prognostic value in chronic hemodialysis patients. The

**Table 1. Baseline characteristics.**

| Characteristics | Value |
|---|---|
| Age (years) | 68.7 ± 12.3 |
| Female, n (%) | 17 (35) |
| BMI (kg/m$^2$) | 26.5 ± 6.3 |
| Diabetes, n (%) | 16 (33) |
| Dialysis vintage (months) | 38 [15–58] |
| Time dialysis session (hours) | 4.1 ± 0.3 |
| Kt/V | 1.8 ± 0.3 |
| Urea (mmol/l) | 21.0 ± 6.4 |
| Creatinine (μmol/l) | 708 ± 172 |
| Bicarbonate (mmol/l) | 23.8 ± 2.4 |
| Plasma proteins (g/l) | 68.0 ± 5.6 |
| Hemoglobin (g/dl) | 12.0 ± 1.2 |
| Parathyroid hormone (ng/l) | 484 [168–608] |
| Vitamin D (μg/l) | 34.1 ± 16.5 |
| Transferrin saturation coefficient, % | 30.3 ± 8.8 |
| Ferritin (ng/L) | 474 ± 255 |
| CRP (mg/l) | 6.6 ± 7.9 |

BMI, body mass index; CRP, C-reactive protein.

baseline value of c20S proteasome was poorly but significantly positively related with LTI and increased c20S proteasome was predictive of muscle mass loss at two years. However, in this study c20S proteasome level did not seem predictive of patient outcome.

The presence of c20S proteasome in blood plasma was first reported by *Wada et al* in 1993 [20]. In the early 2000s its detection with ELISA was developed [21]. Since then c20S proteasome has been studied in several diseases. Elevated c20S proteasome is found in different solid tumors [22], hematological disease [23], critically ill patients [24], autoimmune diseases [25]

**Table 2. Baseline values of nutritional markers.**

| | Nutritional markers | Mean value |
|---|---|---|
| **Biochemical markers** | Albumin (g/l) | 35.7 ± 3.4 |
| | Prealbumin (g/l) | 0.3 ± 0.1 |
| **Body mass** | BMI (kg/m$^2$) | 26.5 ± 6.3 |
| | Triceps skinfold thickness (mm) | 16.0 ± 9.4 |
| | FTI (kg/m$^2$) | 14.9 ± 6.7 |
| | ATM (kg) | 42.0 ± 17.9 |
| **Muscle mass** | Mid-arm circumference (cm) | 31.1 ± 5.9 |
| | Mid-arm muscle circumference (cm) | 25.5 ± 5.8 |
| | Creatinine index (mg/kg/day) | 19.1 ± 2.1 |
| | LTI (kg/m$^2$) | 12.7 ± 2.5 |
| | LTM (kg) | 35.5 ± 8.7 |
| **Dietary intakes** | nPCR (g/kg/day) | 1.1 ± 0.3 |

ATM, adipose tissue mass; BMI, body mass index; FTI, fat tissue index; LTI, lean tissue index; LTM, lean tissue mass; nPCR, normalized protein catabolic rate; PEW, protein energy wasting; PINI, prognostic inflammatory and nutritional index.

**Table 3. Pearson correlation coefficient between c20S Proteasome at baseline and nutritional markers.**

| | Baseline value | 1 year Variation | 2 years Variation |
|---|---|---|---|
| Dry weight | 0.14 | 0.03 | 0.08 |
| BMI | 0.11 | 0.00 | -0.04 |
| LTI | **0.43***  | -0.06 | **-0.56*** |
| LTM | **0.39***  | 0.34 | **-0.54*** |
| FTI | 0.20 | 0.06 | **0.40*** |
| ATM | 0.22 | 0.22 | 0.28 |
| Creatinine Index | 0.14 | -0.19 | 0.11 |
| Albumin | -0.10 | 0.24 | 0.12 |
| Prealbumin | 0.17 | 0.23 | 0.18 |

ATM, adipose tissue mass; BMI, body mass index;; FTI, fat tissue index; LTI, Lean tissue index; LTM, lean tissue mass

* p<0.05.

and surgery [26] where it can be predictive of disease severity and/or patient outcomes. C20S proteasome is probably not specific to muscle mass or protein wasting and could reflect various mechanisms. The origin (white blood, cancer or injured cells) and role of the circulating proteasome are incompletely elucidated and need to be explored.

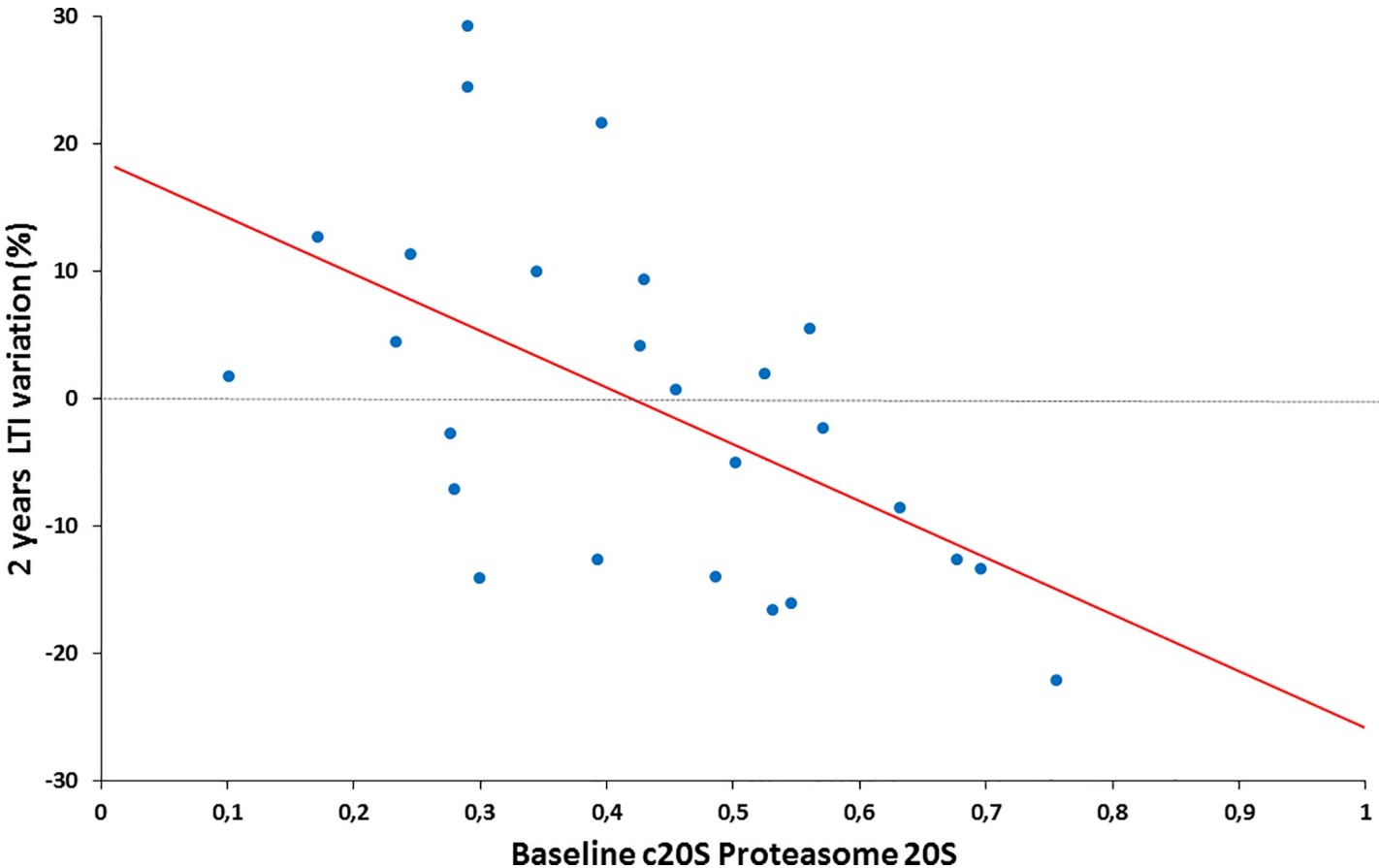

**Fig 2. Linear regression between two years LTI variation and baseline c20S proteasome.** c20S proteasome, circulating 20S proteasome; LTI, lean tissue index measured by multifrequence bioimpedancemetry.

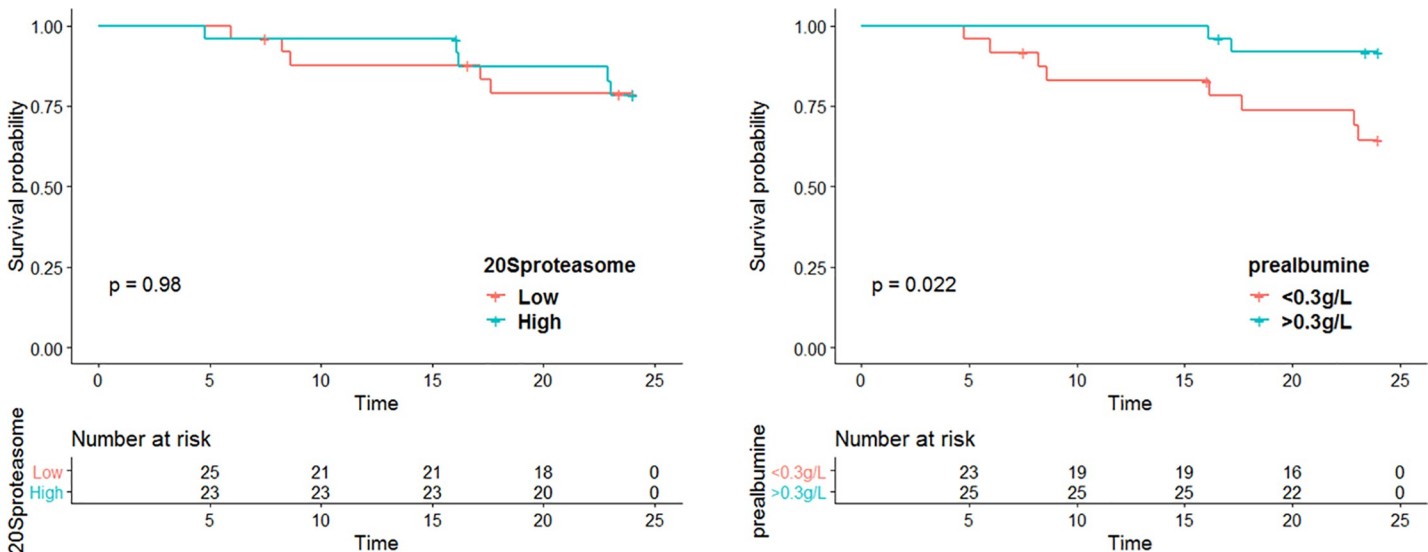

**Fig 3. Two-year survival according to baseline c20S proteasome and prealbumin value.** c20S proteasome, circulating 20S proteasome.

C20S proteasome levels in our study were quite low compared to those in the study of *Fukasawa et al* [13]: values of c20S proteasome level dosage, 1.34 ± 1.12 μg / mL (ELISA) and 1.33 ± 0.53 μg / mL (Western Blot) for CKD patients in the Japanese study, and 0.41 ± 0.31 (ELISA) in our study. The low c20S proteasome levels observed in our study cannot be explained by differences in assay methods since we used the same ELISA assay which had been correlated with Western Blot measurements. A second assay of the 20S proteasome was carried out in the remaining patients at the end of the 2 years of follow-up: it also found lower values than in the study of *Fukasawa et al* (0.6 ±0.16), which makes the assumption of an assay error unlikely.

In the study of *Fukasawa et al*, a negative correlation was found between c20S proteasome levels and abdominal muscle area (AMA) measured in axial CT images at the level of the third lumber spine. In contrast, in the present study we found a low but significant correlation between c20S proteasome levels and muscle mass measured by multifrequence bioimpedancemetry. It could be assumed that this was due to a different technique for measuring muscle mass but the measurement of the muscle area in the lumbar section and bioimpedancemetry are both fairly representative of muscle mass assessed by dual energy X-ray absorptiometry, which is the reference technique [27, 28]. In addition, in our study LTI and c20S proteasome assays were performed the same week. In the study of *Fukasawa et al*, the CT scan used for AMA measurement was not performed during the same week and the indication of this exam was not specified. In fact, the correlation coefficient reported was very low (R = 0.26 in univariate analysis and R = 0.19 in multivariate analysis). Hence, even if the relationship is statistically significant it is not clinically relevant. Moreover, as in our study, no correlation was found between the levels of c20S proteasome and other parameters conventionally used to assess nutritional status such as albumin level, creatinine, nPCR, BMI, IL-6, subcutaneous and visceral abdominal fat mass. In our study, the c20S proteasome levels were not related to decreased muscle mass or PEW criteria at baseline. Unexpectedly, we found a positive correlation between c20 proteasome and LTI at baseline whereas increased proteasome 20S was significantly related to a decrease in LTI at 2 years. Provided that the circulating proteasome originates from the muscle cell, this can be explained by the fact that the quantity released

depends not only on the state of activation of the proteasome within each muscle fiber (i.e. muscle catabolic process) but also on the number of fibers (i.e. muscle mass).

Other new biological markers of muscle have been recently reported notably post-translationally modified muscle-specific ubiquitin ligases and miRNA [29]. In hemodialysis patients, myostatin and IGF1 have been shown to be correlated with HGS and MMI (muscle mass divided by body surface area measured by multifrequence impedancemetry). However, the correlation coefficients were low (R $=$ 0.37 and 0.46, respectively) for HGS and MMI. Moreover, predialysis creatinine performed better. These parameters were associated with first-year mortality in univariate analysis but their effect was non-significant when adjusted for prealbumin [30]. Similar results were obtained in a subsequent study [31]. The use of a single marker may not be sufficient. A strategy based on the identification of a group of biomarkers could be more effective. Niewczas et al have recently shown that proteomic profiling of circulating protein identified 17 proteins enriched for TNF Receptor Superfamily members and associated with a 10-years risk of end stage-renal disease in diabetic patients [32]. Given the role of inflammation in PEW and muscle atrophy it can be assumed that such an approach would be relevant to discover biomarkers associated with the risk of muscle mass loss.

This study has certain limitations. First, with regard to the definition of PEW on the basis of the ISRNM score, we had to adapt to the data available to arrive as close as possible to the initial definition. It is therefore not a fully validated score. Second, the limited number of patients included and the rate of loss of follow-up resulted in a lack of power so that results at 2 years should be viewed with caution. The power of the study was calculated using the seqtest R package. Thus the power to detect a clinically relevant correlation ($R^2 > 0.5$) a type-I error set at 5% was 98.3% at inclusion, 97% at 1 year and 89% at two years. In addition, variability in the dosage of c20S proteasome in the hemodialysis population was very low and made it more difficult to demonstrate a correlation. Finally, overall, the nutritional markers varied little during follow-up. CKD-induced muscle atrophy is a very progressive process and it is possible that muscle mass will stabilize after a period of decrease. Also, although the measurement of c20S proteasome seems of little interest in CKD patients, it could be so in cases of acute kidney injury, where UPS activation has been observed in animal models [10].

In conclusion, c20S proteasome measurement does not seem to be of great value in assessing and predicting the evolution of nutritional status in hemodialysis patients. Muscle atrophy is the result of profound changes in the metabolism of muscle fiber caused by kidney disease. A simultaneous study of the expression of genes in the blood and muscles at the transcriptomic and / or proteomic level in catabolic situations could perhaps identify the best candidates. A biological marker able to predict early the onset of muscle atrophy before its establishment remains to be discovered.

## Supporting information

**S1 Data.**
(XLSX)

## Acknowledgments

We are indebted to Mr. Jeffrey Watts for assistance in the preparation of the manuscript.

## Author Contributions

**Conceptualization:** Julien Aniort.

**Data curation:** Julien Aniort, Marine Freist, Aurélien Piraud, Elodie Gentes.

**Formal analysis:** Julien Aniort, Bruno Pereira.

**Investigation:** Julien Aniort, Carole Philipponnet, Mohamed Hadj Abdelkader, Anne-Elisabeth Heng.

**Methodology:** Bruno Pereira, Anne-Elisabeth Heng.

**Validation:** Julien Aniort, Carole Philipponnet, Cyril Garrouste.

**Writing – original draft:** Julien Aniort, Marine Freist, Bruno Pereira, Anne-Elisabeth Heng.

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
