## [Decision Letter · Decision Letter 0]

17 Jun 2020

PONE-D-20-07087

Circulating 20S proteasome for assessing protein energy wasting syndrome in hemodialysis patients

PLOS ONE

Dear Dr. Aniort,

Thank you for submitting your manuscript to PLOS ONE. After careful consideration, we feel that it has merit but does not fully meet PLOS ONE’s publication criteria as it currently stands. Therefore, we invite you to submit a revised version of the manuscript that addresses the points raised during the review process.

We look forward to receiving your revised manuscript.

Kind regards,

Paolo Fiorina, MD, PhD

Academic Editor

PLOS ONE

Journal Requirements:

Reviewers' comments:

Reviewer's Responses to Questions

**Comments to the Author**

1. Is the manuscript technically sound, and do the data support the conclusions?

Reviewer #1: Yes

Reviewer #2: Yes

2. Has the statistical analysis been performed appropriately and rigorously? 

Reviewer #1: Yes

Reviewer #2: Yes

3. Have the authors made all data underlying the findings in their manuscript fully available?

Reviewer #1: Yes

Reviewer #2: Yes

4. Is the manuscript presented in an intelligible fashion and written in standard English?

Reviewer #1: Yes

Reviewer #2: Yes

5. Review Comments to the Author

Reviewer #1: In this manuscript Aniort et al, describes the role of C20s proteasome assay as a biomarker for the diagnosis of low muscle loss and PEW (protein energy wasting) in hemodialysis patients. For this purpose, they performed an ELISA to measure the 20S subunit in the blood of these patients but authors found out that C20s proteasome levels didn’t correlate with patients’ nutritional markers or clinical outcomes at baseline.

Major flaws

As authors already mentioned in the manuscript, the number of patients is too low.

For this reason, it is difficult to assess the statistic power of this analysis.

I would suggest to increase the number of cases and reperform the statistic.

Minor flaws

In Fig. 1 change “N” with “n” to maintain consistency.

The work done by Aniort et al is of value but need further evidences.

To improve the discussion, I want to suggest other ways to give an explanation on the data of ESRD subjects suffering from both diabetes and PEW. In this case the metabolic disorder of this court of patients might influence the levels of C20s proteasome subunit in some way unlike other patients affected by other comorbidity. In this perspective, an interesting point of view about this topic can be found in this paper (Niewczas M et al, Nat Med, 2019). The evidences contained in this work could be an input to wondering about new biomarkers that could correlate with PEW in a court of ESRD and diabetic patients at least.

Reviewer #2: The Aim of the study was to demonstrate that 20S proteasome assay can predict muscle mass loss and protein energy wasting in hemodialysis patients.

The study is informative and compelling. It provides a better understanding of chronic hemodialysis patients.

6. PLOS authors have the option to publish the peer review history of their article (what does this mean?). If published, this will include your full peer review and any attached files.

Reviewer #1: No

Reviewer #2: No

---

## [Author Response · Author response to Decision Letter 0]

24 Jun 2020

Response to Reviewers

Editor comments

1. We note that you have indicated that data from this study are available upon request. PLOS only allows data to be available upon request if there are legal or ethical restrictions on sharing data publicly. For information on unacceptable data access restrictions, please see http://journals.plos.org/plosone/s/data-availability#loc-unacceptable-data-access-restrictions.

Reply : Because there are no restriction we have upload the minimal anonymized data set necessary to replicate our study finding as Supporting Information files

Reply : Because the data are not a core part of the research being presented in our study we have remaved the phrases that refers to the data as requested (p11 l196 and p12 l216).

Reviewer 1 comments

In this manuscript Aniort et al, describes the role of C20s proteasome assay as a biomarker for the diagnosis of low muscle loss and PEW (protein energy wasting) in hemodialysis patients. For this purpose, they performed an ELISA to measure the 20S subunit in the blood of these patients but authors found out that C20s proteasome levels didn’t correlate with patients’ nutritional markers or clinical outcomes at baseline.

 Major flaws

As authors already mentioned in the manuscript, the number of patients is too low.

For this reason, it is difficult to assess the statistic power of this analysis.

I would suggest to increase the number of cases and reperform the statistic.

Reply: We are aware that the number of patients analyzed is low. Unfortunately this clinical study has been completed to date and we cannot include new patients. Power to detect a correlation between two variables can be calculated. We used the triangular sequential test proposed by Rasch et al. Results were added in the discussion (p15 l285). The number of patients is too low for subgroups or multivariate analysis. However, we believe that we have sufficient statistical power (> 80%) to consider that circulating proteasome is not marker efficient enough to be usable in clinical practice. 

Minor flaws

1. In Fig. 1 change “N” with “n” to maintain consistency.

This was corrected.

2. The work done by Aniort et al is of value but need further evidences. To improve the discussion, I want to suggest other ways to give an explanation on the data of ESRD subjects suffering from both diabetes and PEW. In this case the metabolic disorder of this court of patients might influence the levels of C20s proteasome subunit in some way unlike other patients affected by other comorbidity. In this perspective, an interesting point of view about this topic can be found in this paper (Niewczas M et al, Nat Med, 2019). The evidences contained in this work could be an input to wondering about new biomarkers that could correlate with PEW in a court of ESRD and diabetic patients at least.

Reply : We thank the reviewer for the comment. Indeed proteomic or transcriptomic are interesting ways to discover a panel of biomarkers predictive of muscle atrophy. We have added this reflection and the reference given as an example by reviewer 1 in the discussion part (p14 l274).

 Reviewer #2: 

The Aim of the study was to demonstrate that 20S proteasome assay can predict muscle mass loss and protein energy wasting in hemodialysis patients.

The study is informative and compelling. It provides a better understanding of chronic hemodialysis patients.

.

---

## [Decision Letter · Decision Letter 1]

17 Jul 2020

Circulating 20S proteasome for assessing protein energy wasting syndrome in hemodialysis patients

PONE-D-20-07087R1

Dear Dr. Aniort,

We’re pleased to inform you that your manuscript has been judged scientifically suitable for publication and will be formally accepted for publication once it meets all outstanding technical requirements.

Kind regards,

Paolo Fiorina, MD, PhD

Academic Editor

PLOS ONE

Additional Editor Comments (optional):

Reviewers' comments:

Reviewer's Responses to Questions

**Comments to the Author**

1. If the authors have adequately addressed your comments raised in a previous round of review and you feel that this manuscript is now acceptable for publication, you may indicate that here to bypass the “Comments to the Author” section, enter your conflict of interest statement in the “Confidential to Editor” section, and submit your "Accept" recommendation.

Reviewer #1: All comments have been addressed

Reviewer #2: All comments have been addressed

2. Is the manuscript technically sound, and do the data support the conclusions?

Reviewer #1: Yes

Reviewer #2: Yes

3. Has the statistical analysis been performed appropriately and rigorously? 

Reviewer #1: Yes

Reviewer #2: Yes

4. Have the authors made all data underlying the findings in their manuscript fully available?

Reviewer #1: Yes

Reviewer #2: Yes

5. Is the manuscript presented in an intelligible fashion and written in standard English?

Reviewer #1: Yes

Reviewer #2: Yes

6. Review Comments to the Author

Reviewer #1: (No Response)

Reviewer #2: (No Response)

7. PLOS authors have the option to publish the peer review history of their article (what does this mean?). If published, this will include your full peer review and any attached files.

Reviewer #1: No

Reviewer #2: No

---

## [Editor Report · Acceptance letter]

20 Jul 2020

PONE-D-20-07087R1 

Circulating 20S proteasome for assessing protein energy wasting syndrome in hemodialysis patients 

Dear Dr. Aniort:

I'm pleased to inform you that your manuscript has been deemed suitable for publication in PLOS ONE. Congratulations! Your manuscript is now with our production department. 

Kind regards, 

on behalf of

Dr. Paolo Fiorina 

Academic Editor

PLOS ONE